# The Incidence and Determinants of Metabolic Syndrome Amongst a Group of Migrants to Qatar: A Prospective Longitudinal Observational Cohort Study 24-Months Post-Migration

**DOI:** 10.3390/jcm11010034

**Published:** 2021-12-22

**Authors:** Rana Moustafa Al-Adawi, Kirti Sathyananda Prabhu, Derek Stewart, Cristin Ryan, Hani Abdelaziz, Mohsen Eledrisi, Mohamed Izham Mohamed Ibrahim, Shahab Uddin, Antonella Pia Tonna

**Affiliations:** 1Department of Pharmacy, Hamad General Hospital, Hamad Medical Corporation, Doha 3050, Qatar; rahmed4@hamad.qa; 2School of Pharmacy and Life Sciences, Robert Gordon University, Aberdeen AB10 7GJ, UK; 3Translational Research Institute, Academic Health System, Hamad Medical Corporation, Doha 3050, Qatar; KPrabhu@hamad.qa (K.S.P.); SKhan34@hamad.qa (S.U.); 4College of Pharmacy, QU Health, Qatar University, Doha 2713, Qatar; d.stewart@qu.edu.qa (D.S.); mohamedizham@qu.edu.qa (M.I.M.I.); 5School of Pharmacy and Pharmaceutical Sciences, Trinity College, D02 PN40 Dublin, Ireland; cristin.ryan@tcd.ie; 6Campbellton Regional Hospital Vitalité Health Network, Campbellton, NB E3N 3H3, Canada; alsedace@gmail.com; 7Internal Medicine Department, Hamad General Hospital, Hamad Medical Corporation, Doha 3050, Qatar; MEledrisi@hamad.qa; 8Dermatology Institute, Academic Health System, Hamad Medical Corporation, Doha 3050, Qatar

**Keywords:** metabolic syndrome, migration, incidence, determinants, Qatar

## Abstract

While there is some evidence that migration to Western countries increases metabolic syndrome (MetS) risk, there is a lack of data pertaining to migration to the Middle East. This study aimed to investigate the relationship between migration and MetS incidence following 24-months of residency in Qatar and identify possible MetS determinants. Migrants to Qatar employed at Hamad Medical Corporation (the national health service) aged 18–65 years were invited to participate. Baseline and follow-up screening for MetS included HbA1c, triglycerides, HDL-cholesterol, blood pressure, and waist circumference. MetS-free migrants were rescreened 24-months post-migration, and the World Health Organization STEPwise questionnaire was administered, assessing changes in lifestyle from baseline. Of 1095 migrants contacted, 472 consented to participate, 205 of whom had normal metabolic parameters at baseline; 160 completed follow-up screening. Most participants were males (74.6%, *n* = 153) and Asian (81.0%, *n* = 166/205), and two thirds (66.3%, *n* = 136/205) were nurses. The incidence of new-onset MetS was 17.0% (*n* = 27/160, 95%CI; 11.0–23.0%), with 81.0% (*n* = 129/160, 95%CI; 73.8–86.0%) having at least one MetS element 24-months post-residency in Qatar. Male gender was a risk factor for MetS (adjusted odds ratio (AOR) = 3, *p* = 0.116), as was consuming medication that could induce MetS (AOR = 6.3, *p* < 0.001). There is merit in further research targeting these groups.

## 1. Introduction

The American Heart Association considers metabolic syndrome (MetS) a global epidemic [1,2], affecting around one-quarter of people worldwide and being a significant cause of morbidity and mortality [3,4]. While the syndrome consists of commonly occurring chronic diseases including diabetes mellitus (DM), hypertension (HTN), dyslipidemia, and obesity, it is underdiagnosed by physicians [2,5]. 

In 2009, the International Diabetes Federation (IDF) task force published the Consensus Worldwide Definition of MetS. Any three of the following five elements are sufficient to diagnose MetS; HTN, DM, elevated triglycerides (TG), low high-density lipoprotein cholesterol (HDL-C), or central obesity (Table 1) [6]. In the past decade, the IDF MetS criteria have been the most widely quoted in the literature [7,8,9,10,11,12], and consequently were adopted throughout this research.

In addition to modifiable clinical elements, other factors have been linked to an increased risk of MetS development, including psychological factors of anger, stress, anxiety, and sleep deprivation. [13,14]. Moreover, accumulating evidence has linked migration to MetS and its elements of DM, HTN, dyslipidemia, and central obesity [15,16,17,18,19,20].

The process by which migration increase the risk of MetS is multifactorial, with precipitating factors including urbanisation [21,22], Westernisation [23,24,25,26,27,28], reduced leisure-time physical activity [29], acculturation and acculturation stress [30,31,32,33,34,35], and the migration period. The time spent by migrants away from their original home country before developing MetS or any of its core elements is uncertain. Ranges reported in the literature are between 1 and more than 15 years [15,17,30,36,37,38,39,40]. 

MetS is particularly relevant to the Middle East, with evidence of a high rate of non-communicable diseases amongst migrants to the region. A recent cross-sectional study reported a high prevalence of MetS amongst migrants to Qatar of 48.8% [41]. Moreover, a recent scoping review gave a prevalence of HTN of 30.5%, DM of 9.0–16.0%, and pre-DM of 30.5% amongst South Asian migrants (India, Bangladesh, and Pakistan) [42]. While three studies have assessed MetS prevalence amongst the native Qatar population [41,43,44], only one included migrants, but it did not focus on potential determinants [41].

The aim of this study was to investigate the relationship between migration and the incidence of MetS following 24-month residency in Qatar and to identify possible MetS determinants.

## 2. Materials and Methods

### 2.1. Setting

This study was conducted within the Hamad Medical Corporation (the HMC), the national health service provider in Qatar, which comprises 20 facilities. More than 25,000 employees of considerable ethnic diversity work in the HMC, with employees having migrated from more than 90 countries [45].

### 2.2. Study Design

This study had a prospective, longitudinal, observational design with a nested cross-sectional survey. It was conducted in two phases: the first involved baseline screening of migrants within three months of arrival to Qatar (1 July to 31 December 2017). The second phase was a prospective follow-up of those with normal metabolic parameters at baseline at 24-months post-migration. A summary of the recruitment and research processes is shown in Figure 1.

### 2.3. Sample Size

Literature MetS rates amongst migrants range from 17.0–39.0%, with the majority of studies reporting 20.0–30.0% [46,47,48,49,50]. Using an estimated rate of 25.0 ± 5%, the study sample size was calculated to be 289 participants, with a precision of 5.0% and 95.0% CIs [51].

### 2.4. Study Participants and Recruitment

The study population comprised migrants employed at HMC, where all new employees are subjected to a pre-employment medical examination. Participants were included if they were new migrants and had joined HMC during the 6-month recruitment period (July–December 2017) and were aged 16–65 years. Participants were excluded if they were pregnant females or former citizens in Qatar or the Gulf region, due to cultural and economic similarities between the Gulf countries.

Phase 1 (screening): Following ethical approval, the pre-employment staff clinic provided contact details for those who joined HMC from 1 July to 31 December 2017. Participants who fulfilled the inclusion criteria were contacted via telephone and provided with brief study information. If interested, an appointment was booked at which further information was provided and written consent obtained.

Phase 2 (follow-up): Study participants were those identified as MetS-free at Phase 1. Those identified as having MetS or any MetS elements at Phases 1 or 2 were referred for appropriate management and follow-up.

### 2.5. Data Collection Tools and Data Collection

A data collection tool was developed to address the research objectives, and was piloted among 10 participants to ensure accuracy and comprehensiveness. The pilot data was excluded from the main study. The following data were extracted from the electronic medical records at baseline and follow-up: socio-demographic information, physical measurements, and biochemical measurements of blood glucose and lipid profiles (TG and HDL-C). Data collection was repeated in a 50% random sample by a second researcher to confirm reliability.

At follow-up, participants were contacted to remind them to attend any HMC laboratory, and to book an appointment to complete the WHO STEPwise questionnaire. Laboratory results were extracted from electronic medical profiles, and reliability was again confirmed. The follow-up laboratory tests were fasting TG, fasting blood glucose (FBG), HDL-C, and HbA1c.

The WHO STEPwise characterised any changes in lifestyle as part of the migration process. This validated questionnaire was developed to aid the surveillance of risk factor trends within and between countries [52] and comprises eleven core domains. Participants were requested to answer each question relating to lifestyle habits at baseline and 24-months post-migration.

### 2.6. Data Handling and Analysis

Data were entered into Microsoft Excel^®^ (Microsoft Corporation, 2018) then exported to SPSS^®^ Version 26 (IBM Corp. Released 2015. IBM statistical analysis for Windows Version 26. Armonk, NY: USA IBM Corp). The Kolmogorov–Smirnov test determined the normality of continuous data. The incidence of new MetS-development and each element were determined by Incidence rate = (Number of new MetS cases or elements)/Number of participants who completed the follow-up) × 100 [53]. Differences in means of normally distributed data from baseline to follow-up were tested using the paired sample *t*-test. The Wilcoxon Signed Rank test was used for non-normally distributed data. Associations between the incidence of MetS or the elements (1 or 2 elements) with the categorical parameters and the non-normally distributed continuous parameters were tested using the Pearson Chi-square test. Associations with the normally distributed parameters were tested using one-way ANOVA. Significant variables from the univariate analysis were utilised to construct the multiple univariate logistic regression model to determine any association with MetS incidence. All *p*-values presented were two-tailed, and *p*-values < 0.05 were considered statistically significant.

### 2.7. Research Ethics

The study was approved by the School of Pharmacy and Life Sciences ethics committee at Robert Gordon University (RGU) and the Institutional Review Board (IRB) of the Medical Research Center at the HMC in Qatar [RGU S97; HMC-IRB Registration: SCH-HMC-020-2015]. Written consent was obtained from all participants.

## 3. Results

During the study period, 1379 employees joined the HMC, of whom 1084 were contacted (Figure 2). Following access to individual electronic medical profiles, 205 participants were identified as having no element of MetS and were included in the follow-up. Seven participants became pregnant and were excluded from follow-up, and 38 only partially completed the laboratory follow-up (Figure 2).

Of the 205 MetS-free participants at baseline, 153 (74.6%) were males, and 166 (81.0%) came from Asia. The majority were healthcare providers, predominantly nurses (*n*= 136, 66.3%; Table 2).

Of the 205 eligible participants, 196 (95.6%) completed the STEPwise questionnaire 24-months post-residency in Qatar. Most were non-smokers (*n* = 181, 92.3%) and met the WHO criteria for physical activity (*n* = 183, 93.4%). The lifestyle parameters at baseline and 24-months post-migration are given in Table 3. The mean times spent in vigorous and moderate activities at follow-up dropped by 50% compared to baseline (*p* = 0.001 for both, Wilcoxon Signed Rank test). The reduced moderate and vigorous physical activity levels at follow-up were not associated with MetS incidence (*p* = 0.792, *p* = 0.680 respectively, Pearson Chi-Square test).

At follow-up, 160 of 205 participants completed all the requested laboratory workup (the five elements of MetS). The incidence of new-onset MetS during the 24 months of residing in Qatar was 17.0% (*n* = 27/160, 95% CI; 11.0–23.0%); 81.0% (*n* = 129/160) of participants developed at least one element of MetS (Table 4). Only 19.0% (31/160) of participants were still MetS free 24-months post-migration.

As shown in Table 4, at follow-up more than half of the participants developed central obesity (56.5%, *n* = 108/191, 95% CI; 49.0–64.0%), 33.5% (*n* = 56/167, 95% CI; 26.0–41.0%) developed elevated glycaemic parameters (either abnormal FBG, HbA1c or on DM medications), while 33.0% (*n* = 65/197, 95% CI; 27.0–40.0%) developed HTN (elevated SBP, and DBP, or on antihypertensive medications), and 18.7% (*n* = 31/166, CI; 13.0–26.0%) had new-onset hypertriglyceridemia.

All metabolic parameters were statistically significantly raised at follow-up compared to baseline, with the exception of SBP (Table 5). FBG values were not compared because these were not available for all participants at baseline.

There was a significant association between male gender and developing MetS (Chi-square = 13.4, df = 4, *p* = 0.01). Additionally, administration of medications that potentially induce MetS was significantly associated with an increased risk of MetS or its elements (Chi-square = 16.9, df = 4, *p* = 0.002) compared to those not taking such medications (Table 6). Age was not compared as most participants were of a similar age (mean age 31.2 + 5.2 years). Ethnic groups, marital status, education, occupation, diet, and exercise had no statistically significant effect on incidence of new-onset MetS. Of note, no patients were identified as having five elements of MetS at the end of the 24 months. 

Univariate logistic regression analysis was undertaken to identify the determinants of MetS. Consuming medications that potentially induce MetS (Such as β blocker, steroids and anti-psychotics) was associated with a four-fold higher risk of MetS (unadjusted OR 4.4, 95% CI; 1.74–10.92, *p* = 0.001). Multiple logistic regression analysis was used to adjust for gender and physical activities (moderate and vigorous, baseline, and follow-up). Consuming medications that potentially induce MetS was found to be associated with a six-fold risk of MetS (AOR 6.3, 95% CI; 2.27–17.73, *p* < 0.001; Table 7).

## 4. Discussion

Key study findings were that among migrants with normal metabolic parameters at baseline, 17.0% (95% CI; 11.0–23.0%) developed MetS during their 24-months of residence in Qatar, with around 81.0% (95% CI; 73.8–86.0%) developing at least one element of MetS—most commonly central obesity. Administration of medications that potentially induce MetS increased MetS risk by six-fold.

The incidence of MetS amongst initially MetS-free Qatar migrants rose to 17.0% over the study period. Van der Linden reported similarly high rates of MetS among Ghanaian migrants to Europe, compared to their counterparts in Ghana (ranged from 31.4% to 38.4% vs 8.3%, respectively) [50].

In this study, more than half of the study population developed central obesity during the study, most likely due to reduced physical activity and a diet with fewer fruit and vegetable portions—as highlighted in the analysis of the survey data. In a systematic review of 39 cross-sectional studies, Goulão et al. highlighted the positive association between obesity prevalence and the length-of-stay in a host country [54,55]. Migration was associated with weight gain, which directly increased with the duration of migration. Similar to the findings of the current study, a recent study conducted amongst Qatar migrants in 2020 also highlighted that male migrants were more likely than females to develop MetS [41]. Central and South American male migrants to Washington tended to develop MetS more than female migrants [47]. This variation was attributed to sex hormones that make men more prone to develop central obesity and subsequent insulin resistance [56,57].

Regarding sociodemographic parameters, previous studies have shown the MetS-protective effect of being unmarried (single, divorced or widow), having high education, occupation levels, and consuming diets rich in fruits and vegetables [58,59,60,61,62,63,64,65,66,67]. Other studies have also highlighted the variation of MetS prevalence between ethnic groups [41,68], being most marked in South Asian migrants. Conversely, the current study showed no association between occupation, education, marital status, ethnic origin, and diet with MetS incidence. However, the limited sample size, the extensive medical background (95%, *n* = 195) with similar work environments, and the lack of ethnic variability (81.0% were Asians) could explain these differences.

Compared to migrants not receiving any medications that alter metabolic parameters, migrants consuming these medications were at increased risk of MetS by six-fold. The most commonly reported medications were anti-psychotics, β blockers, steroids, and diuretics. Several previous studies have confirmed these adverse metabolic effects, with proposed mechanisms including the promotion of weight gain, fat disposition in the visceral area, subsequent impaired glucose tolerance, and insulin resistance [69,70,71,72,73,74,75]. Caution should be exercised when prescribing these medications, with periodic monitoring of metabolic parameters [76].

There are numerous strengths to this study. While all previous studies have focused on MetS prevalence at one time-point, this is the first study to screen migrants for MetS at baseline and determine its incidence 24-months post-migration. Hence, the current study associates MetS incidence to the migration process itself and subsequent lifestyle modifications. In addition, this is the first study to assess MetS and its potential determinants amongst migrants to the Middle East and the Gulf. A validated and well-established questionnaire (WHO STEPwise) was used to address lifestyle modifications [77].

There are limitations to this study; hence, the study findings should be interpreted with caution. These include the relatively low participation rate, resulting in the study size being below the calculated sample estimation. There are potential issues of generalisability beyond the study setting. Notably, most of the population were Asian, and almost all had a medical background. Baseline questionnaire data may also be subject to recall bias, and the follow-up period of two years was relatively short. Evidence indicates that a more extended migration period may be associated with higher MetS prevalence [15].

This study provides evidence on the impact of migration to the Middle East in new-onset MetS incidence. Assessment and follow-up of these migrants in terms of MetS and its elements will generally improve health status, control of risk factors, decrease the likelihood of longer-term consequences and enhance quality of life. This study will guide policymakers within the Ministry of Public Health and the HMC in implementing preventative measures to combat MetS among migrants and develop strategies for early warning systems. Further studies with ethnic and occupational diversity and a more extended follow-up period are warranted to determine MetS incidence and its determinants amongst Middle East migrants.

## 5. Conclusions

There is limited evidence concerning the incidence of MetS and its elements post-migration, particularly in the Middle East. Migrants to Qatar, particularly males, showed increased MetS incidence 24-months following migration. Consuming medications that potentially induce MetS was a significant determinant. These factors should be the subject of prospective intervention studies.

## Figures and Tables

**Figure 1 jcm-11-00034-f001:**
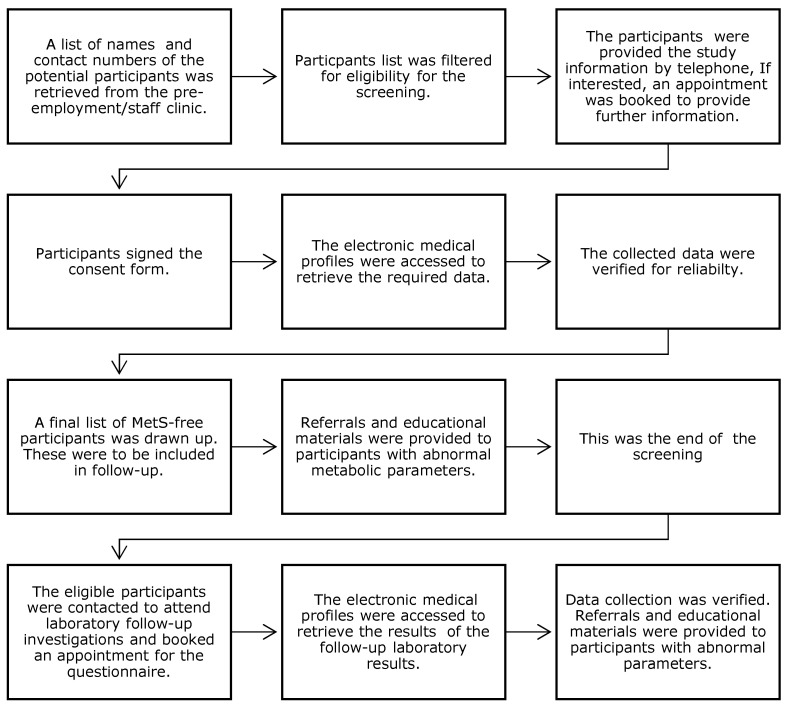
Flow chart summarising the research process in chronological order.

**Figure 2 jcm-11-00034-f002:**
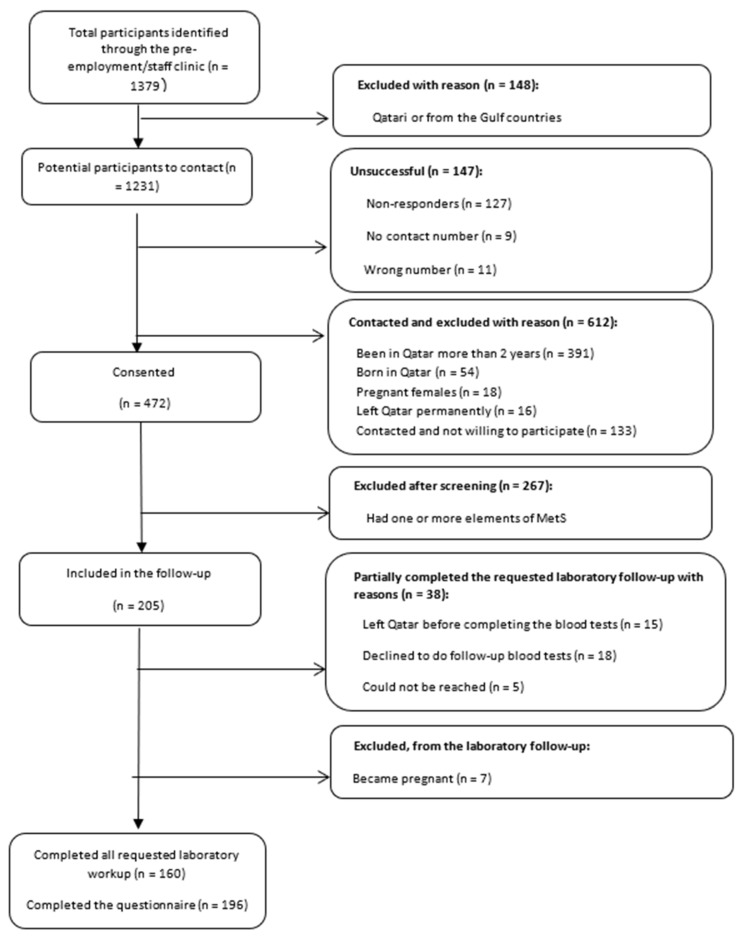
Participant flow at screening follow-up.

**Table 1 jcm-11-00034-t001:** Updated criteria for clinical diagnosis of Metabolic Syndrome [6].

Measure	Categorical Cut Points
Patients were diagnosed as having MetS if they had any three out of the following five elements:	
Elevated TG	≥1.7 mmol/L, or drug treatment for elevated triglycerides
Reduced HDL-C	<1.0 mmol/L in males, <1.3 mmol/L in females, or drug treatment for reduced HDL-C
Elevated BP	Systolic ≥130 and/or diastolic ≥85 mmHg, or on antihypertensive drug treatment
Elevated FBG	≥5.5 mmol/L, or drug treatment for high glucose
Elevated WC	Country/ethnic group waist circumference (a measure of central obesity) [3]
	Europids: Male ≥ 94 cm, Female ≥ 80 cm
	South Asians: Male ≥ 90 cm, Female ≥ 80 cm
	Chinese: Male ≥ 90 cm, Female ≥ 80 cm
	Japanese: Male ≥ 90 cm, Female ≥ 85 cm
	Ethnic South and Central Americans: Use South Asian recommendations until more specific data are available.
	Sub-Saharan Africans: Use European data until more specific data are available.
	Eastern Mediterranean and Middle East (Arab) populations: Use European data until more specific data are available.

TG; triglycerides; HDL-C, high-density lipoprotein cholesterol, BP; blood pressure; FBG, fasting blood glucose; WC, waist circumference.

**Table 2 jcm-11-00034-t002:** Characteristics of the MetS-free participants (*n* = 205) at baseline within 3 months of arrival to Qatar in 2017.

Characteristic	Values % (*n*)
Age	31.2 ± 5.2 years ^¥^
Gender	
Male	74.6% (153)
Female	25.4% (52)
Ethnic origin	
Arabs	8.8% (18)
Asian	81.0% (166)
Africans	2.4% (5)
Europeans	3.4% (7)
Others *	4.4% (9)
Marital status	
Married	48.8% (100)
Single	50.2% (103)
Divorcee	1.0% (2)
Education	
Below bachelor’s degree	1.5% (3)
Graduate (Bachelor & Diploma)	77.0% (158)
Postgraduate (Master and above)	21.5% (44)
Occupation	
Doctors	13.7% (28)
Nurses	66.3% (136)
Allied healthcare	15.1% (31)
Others **	4.9% (10)

WHO, World Health Organisation. ^¥^ ± represents standard deviation (SD) of the mean. * Others included: North Americans, South Americans, Australians–Oceanians. ** Other occupations included: clerk, engineer, IT and aides. Median and IQ range reported due to data skewness.

**Table 3 jcm-11-00034-t003:** Lifestyle parameters at baseline and 24-months post-migration (*n* = 196).

Lifestyle Parameters	Baseline % (*n*)	24-Months Post-Migration % (*n*)
Smokers		
Yes	2.1% (4)	11.2% (22)
No	92.3% (181)	88.8% (174)
Missing	5.6% (11)	0
Alcohol consumers		
Yes		41.8% (82)
No	NA	58.2% (114)
Mean (SD) number of days fruit consumed in a week (days/week)	3.8 ± 2.3 days/week ^¥^	4.4 ± 2.3 days/week ^¥^
Mean (SD) number of servings of fruit consumed on average per day	1.32 ± 0.7 serves/day ^¥^	1.25 ± 0.6 serves/day ^¥^
Mean (SD) number of days vegetables consumed in a week (days/week)	4.7 ± 2.2 days/week ^¥^	4 ± 2.3 days/week ^¥^
Mean (SD) number of servings of vegetables consumed on average per day	1.6 ± 0.7 serves/day ^¥^	1.35 ± 0.6 serves/day ^¥^
Median (IQR) number of mins of moderate activities per week	1800 (540–2850) min ^¶^	1410 (418–2400) min ^¶^
Median (IQR) number of mins of vigorous activities week	120 (0–300) min ^¶^	60 (0–240) min
Met WHO criteria for moderate activity		
Yes	89.3% (175)	84.7% (166)
No	9.2% (18)	15.3% (30)
Missing	1.5% (3)	0
Met WHO criteria for vigorous activity		
Yes	54.6% (107)	43.9% (86)
No	42.3% (83)	51.0% (100)
Missing	3.1% (6)	5.1% (10)
Met the WHO criteria for physical activity (vigorous or moderate)		
Yes	93.4% (183)	91.3% (179)
No	6.6% (13)	8.7% (17)
Met WHO recommendations for diet baseline		
Yes	5.1% (10)	2.5% (5)
No	94.9% (186)	97.5% (191)

^¥^ ± represents standard deviation (SD) of the mean. ^¶^ Median and IQ range reported due to data skewness.

**Table 4 jcm-11-00034-t004:** Incidence of MetS and MetS elements 24-months post-migration.

Metabolic/Parameters	Values % (*n*)	95% CI
Zero elements of MetS (*n* = 160) ^¥^	19.0% (31)	14.0–26.0%
One element of MetS (*n* = 160) ^¥^	33.0% (52)	25.0–40.0%
Two elements of MetS (*n* = 160) ^¥^	31.0% (50)	24.0–39.0%
Three or more elements of MetS (*n* = 160) ^¥^	17.0% (27)	11.0–23.0%
HTN		
SBP ≥ 130 mmHg (*n* = 197) ^¥^	19.3% (38)	14.0–26.0%
DBP ≥ 85 mmHg (*n* = 197) ^¥^	13.7% (27)	9.0–19.0%
SBP ≥ 130 mmHg or DBP ≥ 85 mmHg or on antihypertensive medications (*n* = 197) ^¥^	33.0% (65)	27.0–40.0%
DM, pre-DM		
FBG > 5.5 mmol/L or on medications (*n* = 169) ^¥^	26.6% (45)	20.0–34.0%
FBG > 5.5 mmol/L or HbA1c > 5.6% or on medications (*n* = 167) ^¥^	33.5% (56)	26.0–41.0%
FBG > 5.5 mmol/L or HbA1c > 6.5% or on medications (*n* = 167) ^¥^	3.6% (6)	1.0–8.0%
Dyslipidaemia		
TG ≥ 1.7 mmol/L (*n* = 166) ^¥^	18.7% (31)	13.0–26.0%
HDL-C < 1.03 mmol/L for males or <1.29 mmol/L for females (*n* = 165) ^¥^	9.7% (16)	6.0–15.0%
HDL-C < 1.03 mmol/L for males or <1.29 mmol/L for females or TG ≥ 1.7 mmol/L or on medications (*n* = 166) ^¥^	30.7% (51)	22.0–37.0%
Obesity		
WC ≥ 90 cm for male and ≥80 cm for females * (*n* = 191) ^¥^	56.5% (108)	49.0–64.0%
WC above the range * or on anti-obesity medications (*n* = 192) ^¥^	56.8% (109)	49.0–64.0%

MetS, metabolic syndrome; HTN, hypertension; SBP, systolic blood pressure; DBP, diastolic blood pressure; DM, diabetes mellitus; FBG, fasting blood glucose; HbA1c, glycated haemoglobin; TG; triglycerides; HDL-C, high-density lipoprotein cholesterol; WC, waist circumference. * WC ≥ 90 cm for males and ≥80 cm for females, except for Europid males ≥ 95 cm and Japanese females ≥ 85 cm—please refer to Table 1. ^¥^ The number of the participants vary between the parameters, due to missing data. Thirty-eight partially completed the laboratory workup, while 160 completed it all.

**Table 5 jcm-11-00034-t005:** Comparison of metabolic parameters at baseline and follow-up.

Metabolic Parameter	Baseline (2017) ^¥^	Follow-Up (2019) ^¥^	*t*-Value ^¶^	df	*p*-Value *
SBP mmHg (*n* = 196)	119 ± 9.3	120 ± 12.0	−1.73	195	0.085
DBP mmHg (*n* = 197)	71 ± 9.0	75 ± 10.0	−4.90	196	<0.001
HbA1c % (*n* = 166)	5.1 ± 0.3	5.4 ± 0.3	−13.90	165	<0.001
TG mg/dL (*n* = 166)	1 ± 0.4	1.2 ± 0.7	−4.56	165	<0.001
HDL-C mg/dL (*n* = 165)	1.29 ± 0.3	1.34 ± 0.3	−3.15	164	0.002
WC cm (*n* = 192)	84.2 ± 5.5	88.4 ± 6.9	−12.05	191	<0.001

Abbreviations: SBP, systolic blood pressure; DBP, diastolic blood pressure; HbA1c, glycated haemoglobin; TG, triglycerides; HDL-C, high-density lipoprotein cholesterol. ^¥^ ± represents standard deviation (SD) of the mean. ^¶^ Paired Sample t-test test-value. * Paired Sample t-test.

**Table 6 jcm-11-00034-t006:** Comparison of migrants with one, two, three and four elements of MetS and MetS-free migrants 24 months post-migration to Qatar regarding variables of demographics and the modifiable risk factors (lifestyle variables) (*n* = 160) ^α^.

	No MetS% (*n*)	One Element of MetS % (*n*)	Two Elements of MetS % (*n*)	Three Elements of MetS % (*n*)	Four Elements of MetS % (*n*)	Test Value	df	*p*-Value
Gender								
Male	64.5% (20)	73.1% (38)	86.0% (43)	100% (18)	55.6% (5)			
Female	35.5% (11)	26.9% (14)	14.0% (7)	0	44.4% (4)	13.4	4	0.01 *
Ethnic group								
Arabs	6.5% (2)	7.7% (4)	10.0% (5)	5.6% (1)	0			
Asians	90.3% (28)	78.8% (41)	74.0% (37)	94.4% (17)	88.9% (8)			
Africans	0	1.9% (1)	6.0% (3)	0	0			
European	0	5.8% (3)	4.0% (2)	0	0			
Others ^†^	3.2% (1)	5.8% (3)	6.0% (3)	0	11.1% (1)	11.3	16	0.788 *
Marital status								
Married	22.6% (7)	48.1% (25)	50.0% (25)	61.1% (11)	55.6% (5)			
Single	71.0% (22)	50.0% (26)	50.0% (25)	38.9% (7)	44.4% (4)			
Divorced	6.4% (2)	1.9% (1)	0	0	0	10.2	8	0.254 *
Education								
Below Bachelor’s degree	6.4% (2)	0	0	0	0			
Graduate (Bachelor, Diploma)	83.9% (26)	80.8% (42)	72.0% (36)	77.8% (14)	66.7% (6)			
Postgraduate (Master and above)	9.7% (3)	19.2% (10)	28.0% (14)	22.2% (4)	33.3% (3)	12.7	8	0.123 *
Occupation								
Doctors	6.5% (2)	11.5% (6)	12.0% (6)	11.1% (2)	22.2% (2)			
Nurses	70.9% (22)	71.2% (37)	58.0% (29)	83.3% (15)	66.7% (6)			
Allied healthcare providers	16.1% (5)	15.4% (8)	24.0% (12)	0	0			
Others ^‡^	6.5% (2)	1.9% (1)	6.0% (3)	5.6% (1)	11.1% (1)	11.5	12	0.489 *
Medication that might induce MetS								
Yes	19.4% (6)	13.5% (7)	12.0% (6)	27.8% (5)	66.7% (6)			
No	80.6% (25)	86.5% (45)	88.0% (44)	72.2% (13)	33.3% (3)	16.9	4	0.002 *
Number of days fruit consumed in a week ^¥^	4.2 ± 2.3	4.3 ± 2.3	4.3 ± 2.3	4.7 ± 2.3	3.8 ± 2.6	0.26	4	0.905 ^€^
Mean (SD) number of servings of fruit consumed per day ^¥^	1.3 ± 0.8	1.3 ± 0.5	1.2 ± 0.6	1 ± 0.4	1.2 ± 0.4	0.93	4	0.448 ^€^
Number of days vegetables consumed per week ^¥^	3.6 ± 1.9	4 ± 2.2	4 ± 2.5	4.9 ± 2.4	4.1 ± 2.8	0.85	4	0.493 ^€^
Number of servings of vegetables consumed per day ^¥^	1.3 ± 0.6	1.4 ± 0.6	1.3 ± 0.5	1.4 ± 0.5	1.4 ± 0.5	0.39	4	0.811 ^€^
Met WHO recommendation for moderate activity (≥150 min/week)								
Yes	88.2% (30)	86.3% (44)	80.0% (40)	88.2% (15)	88.9% (8)			
No	11.8% (4)	13.7% (7)	20.0% (10)	11.8% (2)	11.1% (1)	4.67	8	0.792 *
Met WHO recommendation for vagarious activity (≥75 min/week)								
Yes	37.5% (12)	48.0% (24)	46.9% (23)	57.9% (11)	30.0% (3)	5.		
No	62.5% (20)	52.0% (26)	53.1% (26)	42.1% (8)	70.0% (7)	70	8	0.680 *
Met WHO criteria for moderate or vagarious activity								
Yes	93.7% (30)	92.3% (48)	86.0% (43)	88.2% (15)	100% (9)			
No	6.3% (2)	7.7% (4)	14.0% (7)	11.8% (2)	0	7.81	8	0.452 *

^α^*n* = 160 since this only includes participants whose full laboratory workup was available. * Pearson Chi-Square test. ^€^ One-way ANOVA test. ^¥^
*±* represents the standard deviation (SD). ^†^ Others included: North Americans, South Americans, Australians–Oceanians. ^‡^ Other occupations included: clerk, engineer, IT and aides.

**Table 7 jcm-11-00034-t007:** Logistic regression analysis for factors associated with MetS development at 24-months post-migration (*n* = 160) ^¥^.

		Univariate Logistic Regression			Multiple Logistic Regression		
		Unadjusted OR	CI	*p*-value	Adjusted OR	CI	*p*-value
Gender							
Male	17.9% (22)	1.4			3		
Female	13.5% (5)	1 (reference)	(0.48–3.98)	0.534	1 (reference)	(0.76–11.85)	0.116
WHO for moderate activities at baseline met (2017)							
Yes	17.5% (25)	2.9			4.4		
No	6.7% (1)	1 (reference)	(0.37–23.60)	0.285	1 (reference)	(0.37–52.74)	0.242
WHO for vigorous activities at baseline met (2017)							
Yes	18.7% (17)	1.5			0.9		
No	13.4% (9)	1(reference)	(0.62–3.56)	0.379	1 (reference)	(0.31–2.87)	0.939
WHO for vigorous activities at follow-up met (2019)							
Yes	17.3% (23)	1.5	(0.40–5.32)	0.561	0.9	(0.21–4.70)	0.996
No	12.5% (3)	1 (reference)			1 (reference)		
WHO for moderate activities at follow-up met (2019)							
Yes	16.4% (12)	0.9	(0.42–2.29)	0.969	1.2	(0.41–3.32)	0.764
No	16.7% (14)	1 (reference)			1 (reference)		
Taking medications that induce MetS							
Yes	36.7% (11)	4.4	(1.74–10.92)	0.001	6.3	(2.27–17.73)	<0.001
No	11.7% (15)	1 (reference)			1 (reference)		

^¥^ The total number of participants included in the analysis equals 160; however, if there were any single missing values in the participants’ data, the whole data set related to that participant was excluded.

## Data Availability

The datasets generated during and/or analysed during the current study are available from the corresponding author upon reasonable request.

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
