# Peer review of "The Incidence and Determinants of Metabolic Syndrome Amongst a Group of Migrants to Qatar: A Prospective Longitudinal Observational Cohort Study 24-Months Post-Migration"

_jcm, 2021, doi:10.3390/jcm11010034_

Round 1

Reviewer 1 Report

The study "  The incidence and determinants of metabolic syndrome amongst a group of migrants to Qatar: A prospective longitudinal observational cohort study 24 months post-migration" indicates the increase of metabolic syndrome traits in migrants to Qatar during 24-months of migration.”

This is a very interesting study since it is necessary to see how population dynamics impact health in general and on a disease that causes high economic and health costs for the current population such as metabolic syndrome.

Bellow, you can read my concerns:

Although Table 1 contains a lot of valuable information it is difficult to find the precise information and to understand the distribution/approach chosen, could the authors rephrase it so that it is more visual and understandable to the reader?

Similar studies (references 42-44) in the native population are mentioned, but, in my opinion, the results of these studies and the discrepancies or similarities in the results of these studies and the present study should also be mentioned.

Line 89: this should be a section called a sample size.

Explain a little more in detail the following sentence: A data collection tool was developed and piloted among 10 participants.

After performing the one-way ANOVA, no posthoc test was performed? Which one?

Inclusion and exclusion criteria should be specified in more detail.

Lines 142-147: This should be included and explained within the exclusion and inclusion criteria for individuals.

Line 299-301: "Conversely, the current study showed no association between occupation, education, marital status, marital status, ethnic origin, and diet with MetS incidence. However, the limited sample size and lack of ethnic variability (81.0% were 301 Asians) could explain these differences."

I disagree with this sentence, most workers are healthcare workers with the same work environment.

Author Response

Manuscript ID: jcm-1455770: title: The incidence and determinants of metabolic syndrome amongst a group of migrants to Qatar: A prospective longitudinal observational cohort study 24 months post-migration

We are very thankful to the reviewers and the editorial board for their positive review and comments. In the revised manuscript, we have addressed all the issues raised by the reviewers. Below, we address, one by one, the specific issues raised by each reviewer and indicate the revisions that have been incorporated in the manuscript to address these issues.

Reviewer# 1:

  1. Although Table 1 contains a lot of valuable information it is difficult to find the precise information and to understand the distribution/approach chosen, could the authors rephrase it so that it is more visual and understandable to the reader?

Authors Response: Thank you for your comment, the text alliance was modified in the table, hope it is clearer now.

  1. Similar studies (references 42-44) in the native population are mentioned, but, in my opinion, the results of these studies and the discrepancies or similarities in the results of these studies and the present study should also be mentioned.

Authors Response: A statement about MetS prevalence amongst migrants to Qatar at line 66-67. The other findings are compared and contrasted in the discussion section when appropriate (line 295, 302).

  1. Line 89: this should be a section called a sample size.

Authors Response: A subheading was added.

  1. Explain a little more in detail the following sentence: A data collection tool was developed and piloted among 10 participants.

Authors Response: Data collection tool and piloting was amended.

  1. After performing the one-way ANOVA, no posthoc test was performed? Which one?

Authors Response: I would like to confirm that no posthoc analysis was performed after the one-way ANOVA test. Overall P-value was found to be statistically insignificant (P>0.05) and therefore it wasn’t required to use any post-hoc tests for pair-wise comparisons.

  1. Inclusion and exclusion criteria should be specified in more detail. Lines 142-147: This should be included and explained within the exclusion and inclusion criteria for individuals.

Authors Response: the inclusion and exclusion criteria were amended.

  1. Line 299-301: "Conversely, the current study showed no association between occupation, education, marital status, marital status, ethnic origin, and diet with MetS incidence. However, the limited sample size and lack of ethnic variability (81.0% were 301 Asians) could explain these differences."

I disagree with this sentence, most workers are healthcare workers with the same work environment.

Authors Response: Thank you for raising this point. The sentence was modified to include “extensive medical background (95%, n=195) with similar work environment”.

 In summary, in the revised manuscript, we have addressed all the issues raised by the.   We hope that you will now find the revised manuscript acceptable for publication in JCM. If you need any further information, please contact us.

Sincerely

Antonella Tonna, PhD

Robert Gordon University,

School of Pharmacy and Life Sciences,

Po Box

AB10 7GJ, Aberdeen, UK.

Reviewer 2 Report

This is an interesting and well-presented manuscript. The authors investigated the relationship between migration and the incidence of Metabolic Syndrome (MS) following 24-month residency in Qatar, and identified factors that may be associated to its development. Following the initial screening, MS-free migrants were rescreened 24 months post-migration: the incidence of new-onset MS was 17%, but interestingly, 81% of the population studied developed at least one of the MS elements. I have a few comments: (1) According to the findings, the development of MS was associated with the consumption of certain categories of medications, including anti-psychotics, diuretics and steroids. I find this particularly interesting since the administration of these drugs is frequently not taken into consideration in the development of the MS. However, although the authors cite references supporting the implication of anti-psychotics in the development of insulin resistance (ref. 69-72), there are no references for the other categories. Please add two references that first investigated insulin resistance in the liver and muscle following corticosteroids (Journal of Clinical Endocrinology & Metabolism, 54: 131-138, 1982, and Biochemical Journal, 321: 707-712, 1997), and diuretics (European Journal of Endocrinology, 139: 118-122, 1998). (2) It will be useful if the authors add a list of abbreviations at the beginning or the end of the paper.  

Author Response

Manuscript ID: jcm-1455770: title: The incidence and determinants of metabolic syndrome amongst a group of migrants to Qatar: A prospective longitudinal observational cohort study 24 months post-migration

Reviewer# 2:

This is an interesting and well-presented manuscript. The authors investigated the relationship between migration and the incidence of Metabolic Syndrome (MS) following 24-month residency in Qatar and identified factors that may be associated to its development. Following the initial screening, MS-free migrants were rescreened 24 months post-migration: the incidence of new-onset MS was 17%, but interestingly, 81% of the population studied developed at least one of the MS elements. I have a few comments:

Authors Response : We thank reviewer for  your positive feedback and mentioning that “This is an interesting and well-presented manuscript”.  Please find our response to each suggestions/comment  point by point as below.

  1. According to the findings, the development of MS was associated with the consumption of certain categories of medications, including anti-psychotics, diuretics and steroids. I find this particularly interesting since the administration of these drugs is frequently not taken into consideration in the development of the MS. However, although the authors cite references supporting the implication of anti-psychotics in the development of insulin resistance (ref. 69-72), there are no references for the other categories. Please add two references that first investigated insulin resistance in the liver and muscle following corticosteroids (Journal of Clinical Endocrinology & Metabolism, 54: 131-138, 1982, and Biochemical Journal, 321: 707-712, 1997), and diuretics (European Journal of Endocrinology, 139: 118-122, 1998).

Authors Response:  We thanks reviewer for suggesting these useful references. The suggested references were added to the text.

  1. It will be useful if the authors add a list of abbreviations at the beginning or the end of the paper.  

Authors Response: Thank you for your comment. It is great idea. A list with the abbreviation is added to the paper as a comment as I am not sure where to add. 

 In summary, in the revised manuscript, we have addressed all the issues raised by the.   We hope that you will now find the revised manuscript acceptable for publication in JCM. If you need any further information, please contact us.

Sincerely

Antonella Tonna, PhD

Robert Gordon University,

School of Pharmacy and Life Sciences,

Po Box

AB10 7GJ, Aberdeen, UK.

Round 2

Reviewer 1 Report

This point was not addressed properly:

Authors Response: the inclusion and exclusion criteria were amended

Where?

This concern was not explained properly:

Similar studies (references 42-44) in the native population are mentioned, but, in my opinion, the results of these studies and the discrepancies or similarities in the results of these studies and the present study should also be mentioned.

Which are the main findings/ discrepancies or similarities in native population reported in references 42-44 compared with the present study population?

Author Response

  1. This point was not addressed properly:

Authors Response: the inclusion and exclusion criteria were amended.

Where?

Authors Response: This has been further clarified as follows in section 2.4

The study population comprised migrants employed at HMC, where all new employees are subjected to a pre-employment medical examination. Participants were included if they were new migrants and had joined HMC during the 6-month recruitment period (July – December 2017) and were aged 16-65 years. Participants were excluded if they were pregnant females or former citizens in Qatar or the Gulf region due to cultural and economic similarities between the Gulf countries.

Phase 1 (screening): Following ethical approval, the pre-employment staff clinic provided contact details for who joined HMC from 01 July to 31 December 2017.New migrants aged 18–65 years and who joined HMC within a 6-month recruitment period were approached. Participants who fulfilled the inclusion criteria were contacted via telephone and provided brief study information. If interested, an appointment was booked at which further information was provided and written consent obtained.

  1. This concern was not explained properly:

Similar studies (references 42-44) in the native population are mentioned, but, in my opinion, the results of these studies and the discrepancies or similarities in the results of these studies and the present study should also be mentioned.

Which are the main findings/ discrepancies or similarities in native population reported in references 42-44 compared with the present study population?

Authors Response: Thank you for this. The reviewer refers to “discrepancies or similarities in the results of these studies and the present study should also be mentioned.” In our opinion, this is not an aspect that may be discussed in the introduction since the results are not yet available to the reader. The differences and similarities are extensively discussed within the discussion section. Examples of this are provided below:

Migration was associated with weight gain, which directly increased with the duration of migration. Similar to the findings of the current study, a recent study conducted amongst Qatar migrants in 2020 also highlighted that male migrants were more likely than female to develop MetS (44).

Other studies have also highlighted the variation of MetS prevalence with ethnic groups (44,68), being most marked in South Asian migrants. Conversely, the current study showed no association between occupation, education, marital status, ethnic origin and diet with MetS incidence. However, the limited sample size, the extensive medical background (95%, n=195) with similar work environments and lack of ethnic variability (81.0% were Asians) could explain these differences.